# Clinical and Genetic Characteristics of Multiple Epiphyseal Dysplasia Type 4

**DOI:** 10.3390/genes13091512

**Published:** 2022-08-24

**Authors:** Tatiana Markova, Vladimir Kenis, Evgenii Melchenko, Aynur Alieva, Tatiana Nagornova, Anna Orlova, Natalya Ogorodova, Olga Shchagina, Alexander Polyakov, Elena Dadali, Sergey Kutsev

**Affiliations:** 1Research Centre for Medical Genetics, Moskvorechye St., 1, 115522 Moscow, Russia; 2H. Turner National Medical Research Center for Children’s Orthopedics and Trauma Surgery, Parkovaya 64-68, 196603 Saint Petersburg, Russia

**Keywords:** multiple epiphyseal dysplasia type 4, *SLC26A2* gene, pathogenic variants, phenotype variability

## Abstract

Multiple epiphyseal dysplasias (MED) are a clinically and genetically heterogeneous group of skeletal dysplasias with a predominant lesion in the epiphyses of tubular bones. Variants in the *SLC26A2* gene cause their autosomal recessive form (rMED or MED type 4). The accumulation of data regarding the genotype–phenotype correlation can help in the diagnosis and proper management of these patients. The aim of this study was to survey the clinical and genetic characteristics of 55 patients with MED type 4 caused by variants in the *SLC26A2* gene. Diagnosis confirmation was carried out by radiography and custom panel sequencing consisting of 166 genes responsible for the development of hereditary skeletal pathology. This was followed by the validation of the identified variants using automated Sanger sequencing (for six patients) and the direct automatic Sanger sequencing of the coding sequence and the adjacent intron regions of the *SLC26A2* gene for 49 patients. Based on the clinical and genetic analysis of our sample of patients, two main MED type 4 phenotypes with early and late clinical manifestations were identified. An early and more severe form of the disease was observed in patients with the c.835C > T variant (p.Arg279Trp), and the late and milder form of the disease was observed in patients with the c.1957T > A variant (p.Cys653Ser) in the homozygous or compound heterozygous state with c.26 + 2T > C. It was also shown that only three pathogenic variants were found in 95.3% of the alleles of Russian patients with MED type 4: c.1957T > A (p.Cys653Ser), c.835C > T (p.Arg279Trp), and c.26 + 2T > C; thus, it can be assumed that the primary analysis of these variants will contribute to the optimal molecular genetic diagnostics of MED type 4.

## 1. Introduction

Multiple epiphyseal dysplasias (MED) are a clinically and genetically heterogeneous group of skeletal dysplasias with a predominant lesion in the epiphyses of tubular bones [1,2]. Common clinical manifestations of MED include arthralgia, gait disturbance as well as joint deformities resulting from abnormal epiphyseal ossification, which then lead to early osteoarthritis [3,4].

To date, seven main genes and encoded proteins responsible for MED are known: cartilage oligomeric matrix protein (*COMP*), matriline-3 (*MATN3*), α 1–3 chains of type IX collagen (*COL9A1*, *COL9A2*, *COL9A3*), sulfate transporter (*SLC26A2*), and calcium-activated nucleotidase-1 (*CANT1*). The first five have autosomal dominant inheritance, while variants in the *SLC26A2* gene cause the autosomal recessive form (rMED or MED type 4) (OMIM: 226900), which demonstrates the milder clinical form within the spectrum of *SLC26A2*-associated skeletal dysplasia [1,5]. The protein product of this gene functions as a transmembrane transporter of sulfate ions into chondrocytes, which is necessary in maintaining the adequate sulfation of cartilage matrix proteoglycans and in the process of endochondral ossification [6,7]. It has been shown that the clinical severity of the phenotypes correlates with the residual activity of the sulfate transporter due to the combination of alleles that carry different variants in the SLC26A2 gene [5,8]. To date, more than 60 pathogenic variants in the *SLC26A2* (HGMD) gene are known. The most common variant in patients with MED type 4 in European populations is the homozygous c.835C > T (p.Arg279Trp), which was studied by Ballhausen D. et al. in a group of 18 patients. The “Finnish” variant in the splicing donor site c.−26 + 2T > C and c.1957T > A (p.Cys653Ser) is less common [2,9]. A number of authors have suggested the existence of phenotypic features in MED type 4 patients with different genotypes, and the accumulation of data regarding the genotype–phenotype correlation can help in the diagnosis and proper management of these patients [1,10,11,12]. Thus, the aim of this study is to survey clinical and genetic characteristics in a series of patients with MED type 4 caused by variants in the *SLC26A2* gene.

## 2. Materials and Methods

A comprehensive examination was carried out on 55 Russian pediatric patients who were aged from 6 months to 18 years old, came from 51 unrelated families, and presented with clinical and radiological signs of MED type 4. Short stature, waddling gait, arthralgias and joint contractures without obvious neurological or rheumatological history, brachydactyly and clinodactyly, limited flexion in the second metacarpophalangeal joints, varus foot, and patellar subluxation/dislocation were the most common clinical signs before the referral for radiography was given. Radiographic evidence of the deformities of the proximal femur, including the delayed ossification of the femoral heads, coxa vara, coxa breva, deformities of the proximal femoral epiphyses (flattening, fragmentation, Perthes-like changes), delayed ossification and multilayer structure of the patella, and genua valga with or without patellar subluxation/dislocation were the most important findings that led to the preliminary diagnosis and referral to the genetic testing. To clarify the diagnosis, the following methods were used: genealogical analysis, clinical examination, neurological examination according to the standard technique with an assessment of the psycho-emotional sphere, radiography, and genetic testing. The diagnostic path for genetic testing included 2 modalities: single gene sequencing of the *SLC26A2* gene or panel sequencing (custom panel consisting of 166 genes responsible for the hereditary skeletal pathology, with the subsequent validation of the identified variants using automated Sanger sequencing). In 49 cases, single gene (*SLC26A2*) sequencing was employed, and in 6 cases, panel sequencing was used. The selection process for making the decision (panel or single gene sequencing) included an interdisciplinary team discussion of the clinical and radiological data. The consensus of the experts (geneticists and orthopedic surgeons) defined the further diagnostic path: if the patient′s data were compatible with MED type 4, single gene sequencing followed. If the data were inconsistent with MED type 4 but still resembled skeletal dysplasias, panel sequencing was recommended.

The isolation of genomic DNA was carried out from whole blood using the DNAEasy kit (QiaGen, Germany), according to the manufacturer′s standard protocol. The proband′s DNA was analyzed using the original custom AmpliSeq™ panel for the new generation sequence Ion Torrent S5; the panel included coding gene sequences. Sequencing results were analyzed using a standard automatized algorithm for data analysis. Average coverage for this sample was 80×, with a coverage width of (20×)—≥ 90–94%. The detected variants were named according to nomenclature presented on the http://varnomen.hgvs.org/recommendations/DNA website (assessed on 24 July 2022).

To assess the population frequencies of the identified variants, we used a sample from the 1000 Genome Projects, the ESP6500, and The Genome Aggregation Database v2.1.1 [13,14,15]. To assess the clinical significance of the identified variants, the OMIM database and the HGMD^®^ Professional pathogenic variants database version 2021.3 were used. An assessment of the pathogenicity and causality of genetic variants was carried out in accordance with international recommendations for the interpretation of data obtained by massive parallel sequencing [16].

Automatic Sanger sequencing was carried out using ABIPrism 3500xl Genetic Analyzer (Applied Biosystems, Foster City, CA, USA), according to the manufacturer′s protocol. Primer sequences were chosen according to the *SLC26A2* (NM_000112.4) reference sequence. The primers used in the study are listed below in Table 1.

The study was conducted according to the guidelines of the Declaration of Helsinki and approved by the Institutional Review Board of the Research Center for Medical Genetics, Moscow, Russia (protocol code 2021-3, 12 March 2021). The proband′s parents gave informed consent to the genetic testing and the publication of their child’s anonymized data.

## 3. Results

We observed 55 probands (30 males and 25 females) from 51 unrelated families with clinical and radiological signs of MED type 4. Only three families were noted to be consanguineous.

As a result of molecular genetic analysis, seven pathogenic variants in the *SLC26A2* gene were identified, including two novel variants that were not described in the HGMD database: c.469T > C (p.Ser157Pro) and c.1073C > T (p.Ser358Phe) in the compound heterozygous state in one of the patients.. The spectrum of identified variants in the *SLC26A2* gene of Russian patients with MED type 4 is presented in Table 2.

The most common variant was c.1957T > A (p.Cys653Ser), which was found in 60% of the alleles of patients with MED type 4. It was registered in the homozygous state in 45.5% of the patients, and it was compounded with the “Finnish” variant c.−26 + 2T > C in 11.8% of the patients. The c.835C > T variant (p.Arg279Trp), which is the most common variant outside of Finland, was found in only 19% of the alleles of the Russian patients, and the “Finnish” splice-site variant c.−26 + 2T > C was found in 16.3% of the alleles. (Table 3).

Thus, in Russian patients with MED type 4, three pathogenic variants in the *SLC26A2* gene—c.1957T > A (p.Cys653Ser), c.835C > T (p.Arg279Trp), and c.−26 + 2T > C—were found in 95.3% of alleles.

The clinical and radiological features of 35 patients in the sample are summarized in Table 4.

The body height of most patients at the time of examination was within the normal range or slightly reduced according to the normative data, while the median growth in the total sample was −1.23 SD. In most patients, the body proportions were also normal, and only a few showed a mild shortening of the limbs (Figure 1A). The first symptoms of the disease were noticed in the first year of life and included hip dysplasia (43%) or congenital bilateral varus foot deformities—clubfoot or metatarsus adductus (11%) (Figure 1B).

During the first years of life, some of the patients showed signs of having a waddling gait and difficulty in climbing stairs. At the age of 5–9 years, 57% of the patients regularly complained of arthralgias in the hips and knees. Severe early osteoarthritis with a rheumatoid-like course led to the misdiagnosed juvenile arthritis in these patients, which was followed by the administration of steroids and methotrexate despite negative blood tests for inflammatory biomarkers. Ineffective medical treatment led to the referral to the clinical geneticist or orthopedic surgeon, followed by the confirmation of MED. As the disease progressed, all patients developed contractures in large joints. However, one of the early clinical symptoms that allowed us to suspect type 4 MED in the patients of our sample was the relatively limited flexion in the 2nd metacarpophalangeal joint, which could be clearly seen when the fingers were clenched into a fist (Figure 2).

The pathology of the knee joints primarily involved the femoro-patellar joint, which led to the lateral subluxation or habitual dislocation of the patella, usually at the age of 5–7 years (20%) (Figure 3A). Some patients had mild or moderate scoliosis (49%) and lumbar hyperlordosis (54%). The distinctive radiological sign detected in the majority of patients who had lateral knee radiographs available for the analysis (17 out of 18 patients) was the characteristic pattern of ossification—a multilayer structure of the patellar (Figure 3B). The multilayer patella can also be detected earlier (before ossification occurs) using the ultrasound as the “layering” of the US-signal from the patella, as can be seen in the patient with MED type 4 in the first year of life.

The most prominent radiological manifestations included the deformities of the femoral heads (Figure 4). In most of the cases, the first changes were detected symmetrically on both sides, although in some cases, there was an asymmetry of manifestations, which led to the misdiagnoses of the local pathology of the hip joints: Perthes disease, avascular necrosis of the femoral head, or sequelae of septic arthritis (Figure 5).

Most of the epiphyses of the long tubular bones at a young age were characterized by a decrease in the size of the ossification nuclei, which, as the child grew, was transformed into a decrease in epiphyseal height (flattening and elongation along the line of the metaphysis, its radiographic appearance resembling the mushroom-shaped or half-moon-shaped epiphysis). A widening of the metaphysis of the long tubular bones was also noted. For the proximal femur, this resulted in the widening and shortening of the femoral necks, leading to the so-called cervical coxa vara or the coxa breva. From the biomechanical point of view, this shortening leads to the shortening of the lever arm of the hip abductors (the gluteus medius muscle)—the main frontal stabilizer of the hip joint—and as a result, to the main clinical pattern of gait disturbance in this disease, namely the waddling gait (or Trendelenburg gait) and/or the swaying of the trunk when walking (Duchenne phenomenon). As the child grew, the clinical and radiological picture was accompanied by advanced signs of osteoarthritis of the hip and knee joints, which were seen in all patients of the studied group to some extent during their observation (Figure 6).

We conducted a clinical genetic analysis aimed at establishing the genotype–phenotype correlations in patients with pathogenic variants in the *SLC26A2* gene. It was found that in patients with the c.1957T > A (p.Cys653Ser) homozygous variant, the body height was within the normal ranges (−0.7 SD). Despite the fact that 7 out of 19 patients in this subgroup were surveilled or underwent conservative treatment for hip dysplasia diagnosed by ultrasound or radiographs during the first year of life, the first clinical manifestations (hip and knee arthralgias and contractures) appeared at the age of 5–7 years. More prominent signs of musculoskeletal involvement in this group occurred by the age of 10–12 years.

In patients with compound heterozygous c.1957T > A (p.Cys653Ser) and c.−26 + 2T > C variants, the growth was moderately reduced (from −1.12 to −2.99 SD). In patients with homozygous variant c.1957T > A, progressive joint contractures with early onset osteoarthritis also occurred in late childhood or adolescence, but clinical manifestations were more pronounced. Two patients were operated on for bilateral hip dislocation; one of them developed severe scoliosis, and one patient suffered the bilateral dislocation of the patella. The moderate clinodactyly of the 5th finger and the shortening of 4–5 fingers as well as 4–5 metacarpal and metatarsal bones led to the characteristic picture of brachymetaphalangism and were noticed in all the patients of this group (misdiagnosed as Albright osteodystrophy in one of the patients) (Figure 7).

The phenotype of the only patient with the compound heterozygous variant—c.2144C > T (p.Ala715Val) and c.1957T > A (p.Cys653Ser)—was characterized by the presence of a congenital atypical bilateral foot deformity (metatarsus adductus, initially misinterpreted as clubfoot), which did not require surgical correction. The symptoms included gait disturbance that appeared at the age of 2 years, bilateral hip arthritis at the age of 8 years as well as flexion contracture of the right knee joint due to the dislocation of the patella, which required surgical correction. At the age of 11 years, his height was 135 cm (−1.45 SD), and musculoskeletal involvement included elbow flexion contractures, mild scoliosis, shortening of the 4th–5th fingers and toes, and clinodactyly of the 5th fingers. Radiographic examination revealed the deformity of the femoral heads and the shortening of the femoral necks (coxa breva).

Clinical manifestations in seven patients with the c.835C > T variant (p.Arg279Trp) in the homozygous and compound heterozygous states were characterized by the most severe clinical manifestations occurring in early childhood. The first signs of the disease in three patients were verified at the prenatal US screening in the third trimester of pregnancy and were characterized by bilateral clubfoot (4/7) and limb shortening (6/7). At birth and during early postnatal development, two patients demonstrated unilateral cystic edema of the outer ear (2/7), brachydactyly (7/7), and clinodactyly of the 5th fingers (4/7). The height of the patients was lower in comparison to the general sample and ranged from −1.24 SD to −4.08 SD (in four patients, it was less than 2.0 SD). Surgical treatment included the correction of the clubfoot (3/7) in the first year of life and the open reduction of the patella (1/7) at the age of 5 years. MED type 4 was diagnosed by the age of 3–5 years in all patients, and in one patient whose mother had the same homozygous genotype, diagnosis was made at the age of 6 months.

A unique phenotype was revealed in the 11-year-old girl with the newly identified compound heterozygous variants: c.469T > C (p.Ser157Pro) and c.1073C > T (p.Ser358Phe). Her height was within the normal range; however, the main complaint or the phenotypic and radiological signs were atypical for MED type 4. The uniqueness of the disease in our patient included premature asymmetric synostoses of the growth plates of the long bones, which resulted in severe ulnar clubhand and valgus-recurvatum deformities of the lower limbs. She also did not have a typical multilayer patella on the lateral radiographs of her knee joints.

The clinical and genetic analyses of our group of patients made it possible to identify two main MED type 4 phenotypes: those with early and late clinical manifestations. An early form of the disease was observed in patients with the c.835C > T variant (p.Arg279Trp) in the homozygous or compound heterozygous state. The late form of the disease was observed in patients with the c.1957T > A variant (p.Cys653Ser) in the homozygous state, or in the compound heterozygous state with c.−26 + 2T > C with less pronounced clinical manifestations, which made it difficult to diagnose the disease early and properly. It should be noted that this genetic variant prevailed in the group of patients with MED type 4 examined by our group of geneticists and orthopedic surgeons.

## 4. Discussion

*SLC26A2*-associated skeletal dysplasias include a spectrum of nosological forms with an autosomal recessive mode of inheritance, from perinatally lethal achondrogenesis type IB and atelosteogenesis type II to clinically more severe diastrophic dysplasia and relatively mild MED type 4, which together form a clinical continuum that reflects different variants in the *SLC26A2* gene [1,9,17]. Normal or slightly reduced growth with generally preserved body proportions promotes the absence or the rare appearance of typical symptoms such as the cleft palate, “hitchhiker′s thumb “, or cystic edema of the outer ear, thus making it possible to clinically differentiate MED type 4 from diastrophic dysplasia [10]. The main early clinical symptoms of MED type 4 that were present in more than half of the patients in our series were gait disturbances, hip and knee pain, and history of hip dysplasia in the first year of life. All patients had progressive contractures in the large joints and a contracture of the second metacarpophalangeal joint, while clino- and brachydactyly, metatarsus adductus, and congenital clubfoot were less common. Radiographic signs of MED type 4 included the flattening of the epiphyses of the femoral heads with their mushroom- or-moon-like shape, the widening and shortening of the femoral necks, as well as a multilayered patella with the tendency to lateral subluxation and dislocation.

Currently, the largest subgroup of patients that have genetically confirmed MED type 4 and have presented with the clinical and genetic characteristics available in the literature include those with the c.835C > T homozygote variant (p.Arg279Trp) in their *SLC26A2* gene. The phenotypic features were studied in 28 patients from different populations [5,9,10,18]. It was found that the phenotypic manifestations in patients with this variant include the congenital clubfoot, which appeared in the majority of the patients, and brachydactyly. This observation was confirmed by our study. So far, only six cases of MED type 4 homozygous for the c.1957T > A variant (p.Cys653Ser) in the *SLC26A2* gene have been described in the literature. These patients are characterized by a relatively mild early clinical manifestation but a more severe late impairment of the hip joints that require early total arthroplasty as well as the tendency to recurrent habitual dislocation of the patella [1,11,19]. In the observed series of Russian patients with MED type 4, the homozygous variant c.1957T > A (p.Cys653Ser) in the *SLC26A2* gene was the dominant one. This variant was attributed to the late form in terms of the progression of the disease and on the basis of a milder phenotype with normal growth and without major musculoskeletal deformities. In our opinion, orthopedic surgeons should play a key role in diagnosing this variant of the disease since minor general clinical symptoms and the normal growth of patients do not allow for the presumption of skeletal dysplasia during routine clinical examination. The radiographic symptoms in these patients (late appearance of the ossification nuclei of the femoral heads, decreased height and deformation of the epiphyses, shortening and widening of the femoral necks, etc.) can be considered as the signs of local orthopedic pathology (hip dysplasia, Perthes disease, the sequelae of septic or aseptic necrosis of the femoral head, coxa vara), especially in cases with an asymmetrical or a predominantly unilateral radiographic picture.

In contrast, the genotype with the homozygous c.835C > T variant (p.Arg279Trp) accompanied by a more typical clinical and radiological picture of multiple epiphyseal dysplasia was found in only a few cases in the consanguineous families of the Avars of the Republic of Dagestan and were characterized by the early form of the disease. We did not find any significant phenotypic feature in our patients with the “Finnish allele”, c.−26 + 2T > C, except for the slightly more severe course of the disease and the presence of brachymetaphalangism in the subgroup of six patients with the compound heterozygous variants, c.1957T > A (p.Cys653Ser) and c.−26 + 2T > C.

Several studies have focused on the confirmation of the previously supposed correlation between the severity of the phenotype and the degree of activity of the sulfate transporter *SLC26A2,* with a functional analysis for some common variants in the *SLC26A2* gene [8,20,21,22]. Karniski L.P. evaluated the effect of the mutation on the function of the sulfate transporter in Xenopus laevis oocytes and showed a residual sulfate ion transfer rate of 32% for the c.835C > T (p.Arg279Trp) variant as compared with the wild type, and almost equal to the wild-type *SLC26A2* transport for the c.1957T variant > A (p.Cys653Ser), which could explain the existence of phenotypic differences in MED type 4 patients with these variants [21]. At the same time, in the following study by Karniski L.P published in 2004 on mammalian HEK-293 cells, it was found that both of these variants have significant residual activity of the sulfate transporter—from 39% to 62% [22].

The Finnish variant c.−26 + 2T > C, which was found in the majority of patients in the Finnish population with diastrophic dysplasia in the homozygous state, is classified as “severe” and leads to a low level of mRNA (~5% of wild-type alleles); however, being in a trans position with a rather “mild variant” causes a mild form of diastrophic dysplasia or MED type 4 [23,24]. In our study, this genotype was responsible for the occurrence of MED type 4.

One patient from our sample with a more severe MED type 4 phenotype than patients with the homozygous c.1957T > A (p.Cys653Ser) had a combination of alleles: c.2144C > T (p.Ala715Val) and c.1957T > A (p.Cys653Ser). The clinical and radiological features of this variant were previously described by Czarny-Ratajczak M. in three brothers from a Polish family in 2010. The variant was classified as an intermediate form between diastrophic dysplasia and MED, probably associated with the presence of the “severe” variant c.2144C > T (p.Ala715Val). Indeed, the overlapping phenotypes in the spectrum of *SLC26A2*-associated skeletal dysplasias have been reported in the literature [25,26].

Two patients in our sample had rare genotypes. One of them (a 13-year-old girl) had a combination of the alleles c.835C > T (p.Arg279Trp) and c.1957T > A (p.Cys653Ser), which was characterized by a more pronounced short stature (−4.08 SD), severe brachydactyly, unilateral cystic edema of the outer ear, contractures of the large joints, and severe hip and knee arthritis. The second was due to the newly identified compound heterozygous variant in the *SLC26A2* gene: c.469T > C (p.Ser157Pro); the c.1073C > T (p.Ser358Phe) variant clinically manifested as a late form of MED type 4, which had multiple asymmetric lesions in the growth plates of the long tubular bones and presented with the formation of valgus and recurvatum deformities in the lower limbs and a severe ulnar clubhand, accompanied by normal growth and body proportions. Previously, one of the variants in the same codon with the substitution of serine for threonine (p.Ser157Thr) in the trans position by the described variant c.835C > T (p.Arg279Trp) was found in an adult patient with diastrophic dysplasia [27].

The current approach in the general management of patients with SLS26A2-related skeletal dysplasias, including orthopedic surgery for children with major musculoskeletal involvement, remains empirical and does not account genetic variants [28]. The detailed knowledge of clinical and genetic correlations can influence this approach.

The limitation of this study includes the relatively limited number of patients with the pathogenic variant c.835C > T (p.Arg279Trp) in their genotype, which would allow for a comparative analysis of the phenotypic variability of MED type 4. This is possibly due to regional peculiarities in the prevalence of the identified variants in the SLC26A2 gene and requires the collection of data for further analysis. In the current study, we conducted preliminary clinical and genetic analyses aimed at establishing the features of phenotypic manifestations in patients with various pathogenic variants in the SLC26A2 gene. The aim of this study was to conduct a survey of the basic clinical and genetic characteristics in our series of patients with MED type 4. A more profound data analysis is necessary to be able to establish the direct genotype–phenotype correlations.

The most obvious strength of the study, besides the total number of patients, is the interdisciplinary approach to the diagnostics, including a close collaboration between orthopedic surgeons and geneticists. This approach included a team discussion of the clinical and radiological data and the referral track for the genetic testing (single gene sequencing or panel sequencing) based on the consolidated opinion of the group of experts, which led to the shortening of the diagnostic path and the sparing of resources.

## 5. Conclusions

Based on the clinical and genetic analyses of a series of Russian patients with MED type 4, two clinical phenotypes with early and late onset as well as different disease severities can be delineated. It has been shown that more severe and early manifesting forms are more often found in patients with the c.835C > T variant (p.Arg279Trp), while late and milder forms are found in patients with the homozygous c.1957T > A variant (p.Cys653Ser) or the compound heterozygous variant with c.−26 + 2T > C in the *SLC26A2* gene. At the same time, the milder late variant prevailed in the surveyed group, possibly because of the involvement of orthopedic surgeons focused on skeletal dysplasias and the authors’ generally high expertise in the clinical and radiological differential diagnostics of general orthopedic conditions and skeletal dysplasias. It was also shown that only three pathogenic variants were found in 95.3% of the alleles of Russian patients with MED type 4: c.1957T > A (p.Cys653Ser), c.835C > T (p.Arg279Trp), and c.−26 + 2T > C. It can be assumed that the primary analysis of these variants will contribute to the optimal molecular genetic diagnostics of MED type 4.

## Figures and Tables

**Figure 1 genes-13-01512-f001:**
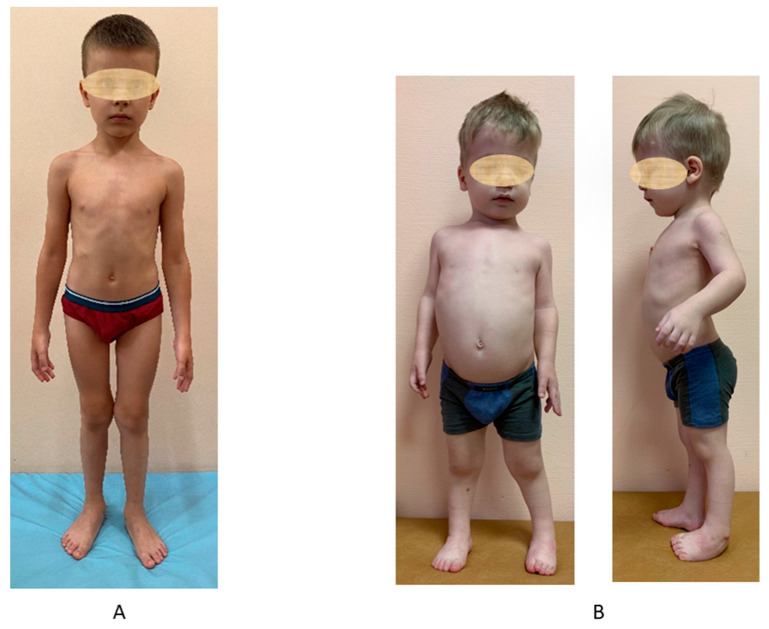
Photos of typical patients with two types of MED type 4. (**A**) A 7-year-old boy with the “late” type c.1957T > A (p.Cys653Ser) homozygous variant—normal body height, mild genua valga, normal hands and feet. (**B**) A 5-year-old boy with the “early” type c.835C > T (p.Arg279Trp) and c.−26 + 2T > C compound heterozygous variant—short stature, genua valga, limited elbow extension, varus feet, brachy- and clinodactyly.

**Figure 2 genes-13-01512-f002:**
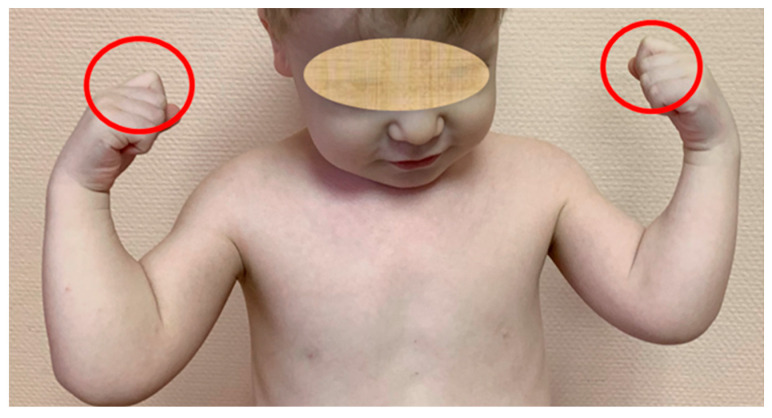
Early clinical hallmark of MED type 4—limited flexion in the second metacarpophalangeal joint, resulting in the “index sign” when the fingers are clenched into a fist (red circles).

**Figure 3 genes-13-01512-f003:**
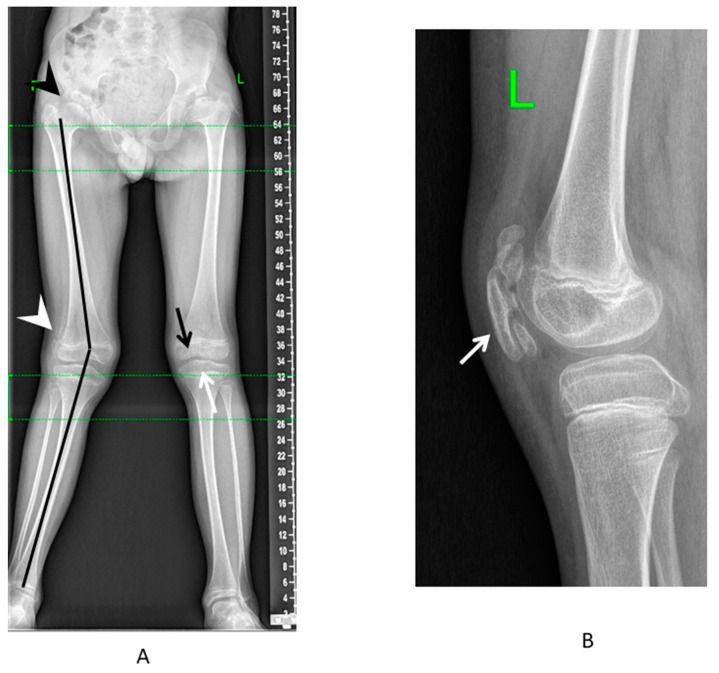
Typical lower limb involvement in patients with MED type 4. (**A**) Standing anteroposterior radiograph of the lower limbs: valgus deformity of the axis (black line), mushroom-shaped deformity of the femoral head (black arrowhead), lateral patellar subluxation (white arrowhead), flattening of the distal femoral (black arrow), and proximal tibial (white arrow) epiphyses. (**B**) Lateral radiograph of the knee—multilayer patella (white arrow).

**Figure 4 genes-13-01512-f004:**
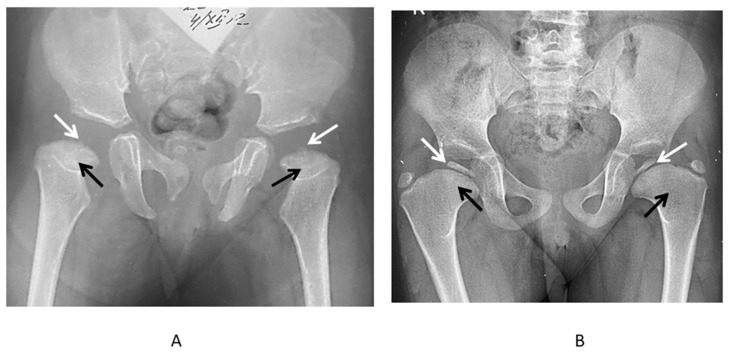
Early radiological changes in the hip joints of the patients with MED type 4. (**A**) Anteroposterior radiograph of the hips of the two 5-year-old patients: remarkably delayed ossification of the epiphyses of the femoral heads (white arrows), shortening of the femoral necks (black arrows). (**B**) Anteroposterior radiograph of the hips of the 6-year-old patient: abnormal ossification (diminished size and flattening) of the epiphyses of the femoral heads (white arrows), shortening of the femoral necks—“cervical coxa vara” (black arrows).

**Figure 5 genes-13-01512-f005:**
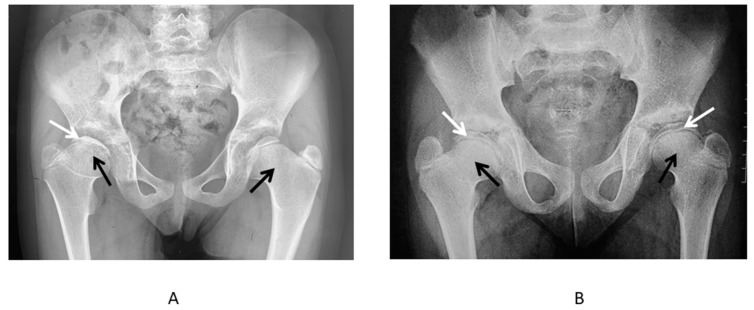
Advanced radiological changes in the hip joints of the patients with MED type 4. (**A**) Anteroposterior radiograph of the hips of the 9-year-old patient: abnormal shape and structure of the epiphyses of the femoral heads—half-moon shape and density resembling that of Perthes disease on the right side (white arrow), shortening of the femoral necks more prominent on the right side (black arrows). (**B**) Anteroposterior radiograph of the hips of the 11-year-old patient: abnormal shape of the epiphyses of the femoral heads—half-moon shape (white arrows), shortening of the femoral necks—the coxa breva (black arrows).

**Figure 6 genes-13-01512-f006:**
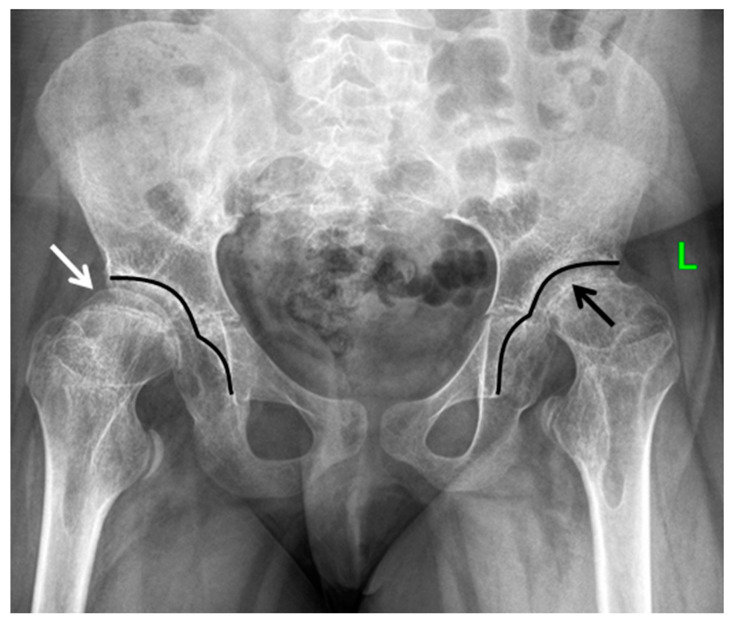
End-stage osteoarthritis of the hip joints of the 15-year-old patient with MED type 4. Anteroposterior radiograph of the hips: abnormal shape and structure of the epiphysis of the femoral head on the right side—half-moon shape (white arrow), secondary reduced size of the epiphysis and narrowing of the joint space on the left side (black arrow), secondary deformity of the acetabulum—doubled “seagull-like” contour (black lines).

**Figure 7 genes-13-01512-f007:**
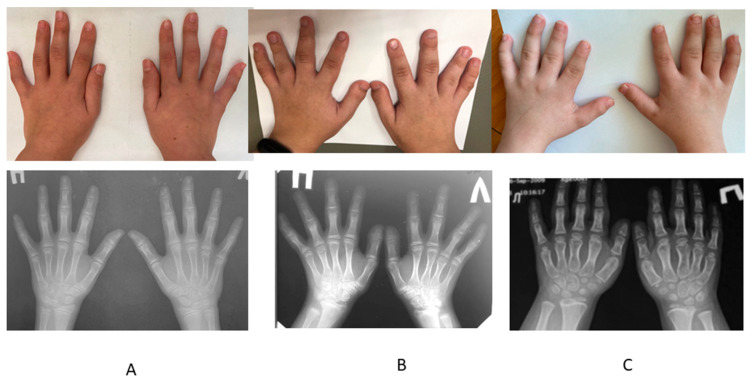
Photos and radiographs of the hands of the patients with different variants in the *SLC26A2* gene. (**A**) Patient with the homozygous c.1957T > A (p.Cys653Ser) variant. (**B**) Patient with the compound heterozygous variant, c.1957T > A (p.Cys653Ser) and c.−26 + 2T > C. (**C**) Patient with the compound heterozygous variant, c.835C > T (p.Arg279Trp) and c.−26 + 2T > C.

**Table 1 genes-13-01512-t001:** Primers used in the study.

Target Gene	Primers	Sequence (5′→3′)
*SLC26A2*	1F	CGAGTTATTGGCTGGTGGTAGC
	1R	CAGCACCGCTCCCTCCCTC
	2F1	GTGAGCACTGAGAATTACTTTATTGATG
	2R1	CTGGGGCACCAATAATATGCC
	2F2	GTGGCTCCCAAAATACGACC
	2R2	GAGATTTCATGTACTAGATTACTTTTC
	3F1	GCTCTGATGATATGTCTCCATGC
	3R1	CCATGTTTCTTGGCAAACATCTC
	3F2	GTGTGGCTGTAGATGCAATAGC
	3R2	GTTTATGTAGTAGAGAGGGGCTAC
	3F3	GTCTTTGAATCTGTGTCTGCTTAC
	3R3	GGAACTGGGAAATGTTGGACAC

**Table 2 genes-13-01512-t002:** The spectrum of identified variants in the *SLC26A2* gene of Russian patients with MED type 4.

*SLC26A2* Variant 1	*SLC26A2* Variant 2	Number of Patients	Reported in
c.1957T > A (p.Cys653Ser)	c.1957T > A (p.Cys653Ser)	25	Mäkitie O., 2003 Hinrichs T., 2010 Gatticchi L., 2021
c.1957T > A (p.Cys653Ser)	c.−26 + 2T > C	13	Kausar M., 2019
c.1957T > A (p.Cys653Ser)	c.2144C > T (p.Ala715Val)	1	Czarny-Ratajczak M., 2010
c.1957T > A (p.Cys653Ser)	1535C > A (p.Thr512Lys)	1	-
c.835C > T (p.Arg279Trp)	1535C > A (p.Thr512Lys)	1	Syvänen J., 2013 Mäkitie O., 2014 Kausar M., 2019 Härkönen H., 2021
c.835C > T (p.Arg279Trp)	c.835C > T (p.Arg279Trp)	7	Superti-Furga A, 1999 Huber C., 2001 Ballhausen D., 2003 Barbosa M., 2010 García M.M., 2014
c.835C > T (p.Arg279Trp)	c.−26 + 2T > C	5	Mäkitie O., 2014 Dwyer E., 2010 Zechi-Ceide RM, 2013
c.835C > T (p.Arg279Trp)	c.1957T > A (p.Cys653Ser)	1	-
c.469T > C (p.Ser157Pro)	c.1073C > T (p.Ser358Phe)	1	-

**Table 3 genes-13-01512-t003:** Frequency of the allelic variants in the *SLC26A2* gene in patients with MED type 4.

*SLC26A2* Allelic Variant	Frequency of the Allelic Variant (*n* = 110)
c.1957T > A	60%
c.835C > T	19%
c.−26 + 2T > C	16.3%
c.1535C > A	1.8%
c.2144C > T	0.9%
c.469T > C	0.9%
c.1073C > T	0.9%

**Table 4 genes-13-01512-t004:** Prevalence and frequency of clinical and radiological features in patients with MED type 4.

Clinical and Radiological Features	Prevalence	Frequency, %
Normal stature (from −2 SD to 2 SD)	28/35	80
Hip dysplasia in the first year of life	15/35	43
Congenital bilateral clubfoot	4/34	11
Arthralgias in the hips and knees	20/35	57
Joint contractures	35/35	100
Brachydactyly and/or clinodactyly	16/35	46
Patellar luxation	7/34	20
Scoliosis	17/35	49
Lumbar lordosis	19/35	54
Multilayer structure of the patella	17/18	94
Deformities of the femoral heads	35/35	100

## Data Availability

Not applicable.

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
