# Peer review of "Clinical and Genetic Characteristics of Multiple Epiphyseal Dysplasia Type 4"

_genes, 2022, doi:10.3390/genes13091512_

Round 1
Reviewer 1 Report
The aim of this study is to survey the clinical and genetic characteristics in the series of patients with MED type 4 caused by variants in the SLC26A2 gene. Based on the clinical and genetic analysis of a series of Russian patients with MED type 4 it was suggested that two clinical phenotypes with early and late onset and different severity of the disease can be delineated. It has been shown that more severe early manifesting forms are most often found in patients with the c.835C>T variant (p.Arg279Trp), and late, milder forms are found in patients with the homozygous c.1957T>A variant (p.Cys653Ser) or compound heterozygous with c. -26+2T>C in the SLC26A2 gene. It was also shown that only three pathogenic variants were found in 95.3% of the alleles of Russian patients with MED type 4: c.1957T>A (p.Cys653Ser), c.835C>T (p.Arg279Trp) and c. - 26+2T>C. It can be assumed that the primary analysis of these variants will contribute to the optimal molecular genetic diagnostics of MED type 4.
The introduction is well structured, highlighting the clinical problem
The material and methods are adequately described and the study can be repeated in another medium. Clinical and radiological criteria that indicate the suspicion of the disease could be included in the methodology The results are clearly expressed with a very demonstrative abundant iconographyA table with the main genotype-phenotype characteristics could be useful
The discussion is correct and adjusted to the results obtained. Limitations of the study and strengths should be included .
Author Response
The authors are grateful to the reviewer for carefully reading the work and making comments. We believe that considering the comments of the reviewer made it possible to significantly improve quality of our manuscript. According to the recommendations of the reviewer, we have summarized the main comments and made the following changes in the "Materials and Methods" section of the manuscript:
- Clinical and radiological criteria that indicate the suspicion of the disease could be included in the methodology
A comprehensive examination of 55 Russian pediatric patients from 51 unrelated families aged from 6 month to 18 years old with clinical and radiological signs of MED type 4 was carried out. Short stature, waddling gait, arthralgias and joint contractures without obvious neurological or rheumatological history, brachydactyly and clinodactyly, limited flexion in the second metacarpophalangeal joints, varus foot, patellar subluxation/dislocation were the most common clinical signs before the referral to the radiography. Radiographic evidence of the deformities of the proximal femur including delayed ossification of the femoral heads, coxa vara, coxa breva, deformities of the proximal femoral epiphyses (flattening, fragmentation, Perthes-like changes), delayed ossification and multilayer structure of the patella, genua valga with or without patellar subluxation/dislocation were the most important findings leading to the preliminary diagnosis and referral to the genetic testing.
We also considered the suggestion of the reviewer regarding the "Results" section:
- A table with the main genotype-phenotype characteristics could be useful
In the study we conducted preliminary clinical and genetic analysis aimed to establishing the features of phenotypic manifestations in patients with various pathogenic variants in the SLC26A2 gene. The aim of this study was to provide survey of the basic clinical and genetic characteristics in our series of patients with MED type 4. The more profound data analysis is necessary for the establishing of the direct genotype-phenotype correlations. We agree with the opinion of the reviewer regarding the usefulness of putting these data into a table, which we will do later, as the sample of patients with different MED type 4 genotypes expands for their reliable comparison. And thanks to the reviewer’s comment we’ll put this explanation into the limitations of the study.
We thank the reviewer for the helpful comment and we have made the following changes to the "Discussion" section:
- Limitations of the study and strengths should be included.
The limitation of the study included relatively limited number of patients with the pathogenic variant c.835C>T (p.Arg279Trp) in the genotype for a comparative analysis of the phenotypic variability of MED type 4, which is possibly due to regional peculiarities in the prevalence of the identified variants in the SLC26A2 gene and requires the collection of data for the further analysis. In the current study we conducted preliminary clinical and genetic analysis aimed to establishing the features of phenotypic manifestations in patients with various pathogenic variants in the SLC26A2 gene. The aim of this study was to provide survey of the basic clinical and genetic characteristics in our series of patients with MED type 4. The more profound data analysis is necessary for the establishing of the direct genotype-phenotype correlations.
The most obvious strength of the study besides the total number of patients is the interdisciplinary approach to the diagnostics including close collaboration between orthopedic surgeons and geneticists. This approach included team discussion of the clinical, radiological data and the referral track for the genetic testing (single gene sequencing or panel sequencing) based on the consolidated opinion of the group of experts which lead to the shortening of the diagnostic path and sparing of the resources.
We are grateful to the reviewer for positive review and helpful comments.
Sincerely,
Collective of authors
Reviewer 2 Report
Review “Clinical and genetic characteristics of multiple epiphyseal dysplasia type 4” by Tatiana Markova et al 2022
Authors of “Clinical and genetic characteristics of multiple epiphyseal dysplasia type 4” described a study with a group of patients which were identified the genetic variants associated with SLC26A2 that can contribute to multiple epiphyseal dysplasia type 4 onset and progression. Authors describe some correlations with specific genetic variants and identified two new variants in the group of patients. However still some questions were raised in the manuscript:
Methods:
Authors mention that have consulted on line “The detected variants were 76 named according to nomenclature presented on the http://varnomen.hgvs.org/recom77 mendations/DNA website” Please add the date of last consultation of the site.
The authors mentioned that “ targeted panel sequencing consisting of 166 genes responsible for the development of hereditary skeletal pathology for 6 patients” But authors did not showed the results for this procedure, or the results showed for this 6 patients were only for SLC26A2 gene analysis?
Authors referenced that “Primer sequences were chosen according to the SLC26A2 (NM_000112.4) reference sequence.” But they did not present the list of primers that were used, its important to provide primers sequences in the manuscript or available in supplementary data.
Results:
Authors referenced that “2 novel variants that were not described in the HGMD data base” , however is not clear if the authors consider a new variant the compound heterozygous c.835C>T (p.Arg279Trp) / c.1957T>A (p.Cys653Ser), because of absence of references. If that is the case, we cannot consider a new variant. Maximum we can say that is a new combination of variants c.835C>T and c.1957T>A as compound heterozygous. Or authors were considering the second combination of mutations c.469T>C (p.Ser157Pro) / c.1073C>T (p.Ser358Phe), which then can be described as new variants. Please clarify in the text.
In table 3 the results described for Prevalence and frequency of clinical and radiological features in the patients with MED type 4 would benefit from the addition of the frequency of each variant for each of the clinical and radiological variants identified in the 35 patients in order to analyze possible impact of variant combination with phenotype.
Discussion
Authors cover the symptoms and correlation between variants and some phenotypes however authors should discuss better what could be the molecular implications of the variants identified and for example why the “Clinical manifestations in 7 patients with the c.835C>T variant (Ñ€.Arg279Trp) in the homozygous and compound-heterozygous state were characterized by the most severe clinical manifestations occurring in early childhood” With could be the molecular implications of Ñ€.Arg279Trp on SLC26A2 that could explain the phenotype. Also if variants alone may not explain the phenotypes, which susceptibility genetic variants could contribute to aggravate the patients phenotype?
Author Response
We thank the reviewer for a detailed analysis of our manuscript. Your comments and suggestions helped us significantly improve the quality of the obtained data analysis and their presentation. In the revised version of this paper, we tried, as much as possible, to consider the comments and suggestions presented in the review. Please find below our responses to each of your comments.
- Authors mention that have consulted on line “The detected variants were named according to nomenclature presented on the http://varnomen.hgvs.org/recom mendations/DNA website” Please add the date of last consultation of the site.
We thank the reviewer for the helpful comment and we have made the following changes to the "Materials and Methods" section:
The detected variants were named according to nomenclature presented on the http://varnomen.hgvs.org/recommendations/DNA website (assesed on 24 July 2022)
- The authors mentioned that “targeted panel sequencing consisting of 166 genes responsible for the development of hereditary skeletal pathology for 6 patients”. But authors did not showed the results for this procedure, or the results showed for this 6 patients were only for SLC26A2 gene analysis?
As recommended by the reviewer, we have made the following changes in the "Materials and Methods" section of the manuscript:
The diagnostic path for genetic testing included 2 modalities: single gene sequencing of SLC26A2 gene or panel sequencing (custom panel consisting of 166 genes responsible for the hereditary skeletal pathology with subsequent validation of the identified variants using automated Sanger sequencing). In 49 cases single gene (SLC26A2) sequencing and in 6 cases – panel sequencing was used. The selection process for the decision making (panel or single gene sequencing) included interdisciplinary team discussion of the clinical and radiological data. The consensus of the experts (geneticists and orthopedic surgeons) defined the further diagnostic path: if the patient’s data was compatible with MED type 4, single gene sequencing followed. If the data was inconsistent with MED type 4 but still resembled to the skeletal dysplasias, panel sequencing was recommended.
- Authors referenced that “Primer sequences were chosen according to the SLC26A2 (NM_000112.4) reference sequence.” But they did not present the list of primers that were used, its important to provide primers sequences in the manuscript or available in supplementary data.
According to the recommendations of the reviewer, we have made the following changes in the "Materials and Methods" section of the manuscript:
Primers used in the study are listed below in Table 1.
Table 1. Primers used in the study.
|
Target gene |
Primers |
Sequence(5'→3') |
|
SLC26A2 |
1F |
CGAGTTATTGGCTGGTGGTAGC |
|
|
1R |
CAGCACCGCTCCCTCCCTC |
|
|
2F1 |
GTGAGCACTGAGAATTACTTTATTGATG |
|
|
2R1 |
CTGGGGCACCAATAATATGCC |
|
|
2F2 |
GTGGCTCCCAAAATACGACC |
|
|
2R2 |
GAGATTTCATGTACTAGATTACTTTTC |
|
|
3F1 |
GCTCTGATGATATGTCTCCATGC |
|
|
3R1 |
CCATGTTTCTTGGCAAACATCTC |
|
|
3F2 |
GTGTGGCTGTAGATGCAATAGC |
|
|
3R2 |
GTTTATGTAGTAGAGAGGGGCTAC |
|
|
3F3 |
GTCTTTGAATCTGTGTCTGCTTAC |
|
|
3R3 |
GGAACTGGGAAATGTTGGACAC |
- Authors referenced that “2 novel variants that were not described in the HGMD data base”, however is not clear if the authors consider a new variant the compound heterozygous c.835C>T (p.Arg279Trp) / c.1957T>A (p.Cys653Ser), because of absence of references. If that is the case, we cannot consider a new variant. Maximum we can say that is a new combination of variants c.835C>T and c.1957T>A as compound heterozygous. Or authors were considering the second combination of mutations c.469T>C (p.Ser157Pro) / c.1073C>T (p.Ser358Phe), which then can be described as new variants. Please clarify in the text.
We considered this recommendation of the reviewer and tried to make additional clarifications in the "Discussions" section:
As a result of molecular genetic analysis, 7 pathogenic variants in the SLC26A2 gene were identified, including 2 novel variants that were not described in the HGMD database: c.469T>C (p.Ser157Pro) и c.1073C>T (p.Ser358Phe) in compound heterozygous state in one of the patients.
- In table 3 the results described for Prevalence and frequency of clinical and radiological features in the patients with MED type 4 would benefit from the addition of the frequency of each variant for each of the clinical and radiological variants identified in the 35 patients in order to analyze possible impact of variant combination with phenotype.
We agree with the reviewer's opinion regarding the possible addition of the frequency of each variant to in order to analyze the possible impact of variant combination with phenotype, which will be presented in our subsequent study and will significantly improve the quality of the obtained data analysis and their presentation.
- Authors cover the symptoms and correlation between variants and some phenotypes however authors should discuss better what could be the molecular implications of the variants identified and for example why the “Clinical manifestations in 7 patients with the c.835C>T variant (Ñ€.Arg279Trp) in the homozygous and compound-heterozygous state were characterized by the most severe clinical manifestations occurring in early childhood” With could be the molecular implications of Ñ€.Arg279Trp on SLC26A2 that could explain the phenotype. Also if variants alone may not explain the phenotypes, which susceptibility genetic variants could contribute to aggravate the patients phenotype?
We agree with the opinion of the reviewer regarding the possible molecular implications of the identified variants on the severity of clinical manifestations. We do not exclude that the existence of phenotypic variability in patients with MED type 4 in our sample is associated with the previously suggested correlation between the severity of the phenotype and the degree of activity of the SLC26A2 sulfate transporter due to different pathogenic variants. In this regard, we present the studies of some authors, which evaluated the effect of common variants in the SLC26A2 gene on the function of the sulfate transporter. Undoubtedly, further research is needed to better understand the molecular mechanisms of phenotypic manifestations.
We are very grateful to the reviewer for detailed analysis and valuable comments, which will help us to improve our manuscript.
Respectfully yours,
Collective of authors.
Round 2
Reviewer 1 Report
The authors have answered the questions. The paper van be accept to publication